# *EMMA*: Generalizing Real-World Robot Manipulation via Generative Visual Transfer

## Abstract

Vision–language–action (VLA) models increasingly rely on diverse training data to achieve robust generalization. However, collecting large-scale real-world robot manipulation data across varied object appearances and environmental conditions remains prohibitively time-consuming and expensive. To overcome this bottleneck, we propose **E**mbodied **M**anipulation **M**edia **A**daptation (*EMMA*), a VLA policy enhancement framework that integrates a generative data engine with an effective training pipeline. We introduce *DreamTransfer*, a diffusion Transformer-based framework for generating multi-view consistent, geometrically grounded embodied manipulation videos. *DreamTransfer* enables text-controlled visual editing of robot videos, transforming foreground, background, and lighting conditions without compromising 3D structure or geometrical plausibility. Furthermore, we explore hybrid training with real and generated data, and introduce *AdaMix*, a hard-sample-aware training strategy that dynamically reweights training batches to focus optimization on perceptually or kinematically challenging samples. Extensive experiments show that videos generated by *DreamTransfer* significantly outperform prior video generation methods in multi-view consistency, geometric fidelity, and text-conditioning accuracy. Crucially, VLAs trained with generated data enable robots to generalize to unseen object categories and novel visual domains using only demonstrations from a single appearance. In real-world robotic manipulation tasks with zero-shot visual domains, our approach achieves over a 200% relative performance gain compared to training on real data alone, and further improves by 13% with *AdaMix*, demonstrating its effectiveness in boosting policy generalization.

## 1 Introduction

Vision–language–action (VLA) models have demonstrated remarkable capabilities in enabling robots to perform complex manipulation tasks from natural language instructions and visual inputs (Black et al., 2024; Intelligence et al., 2025; Brohan et al., 2023; Kim et al., 2024; NVIDIA et al., 2025c; Deng et al., 2025). However, their success critically depends on large-scale, diverse training data. Collecting real-world robot manipulation data through human teleoperation is labor-intensive and expensive, severely limiting the scale and visual diversity of available datasets. While simulation offers a scalable alternative for generating annotated trajectories (Geng et al., 2025; Lin et al., 2024; Mu et al., 2025; Katara et al., 2023; Lin et al., 2024), simulated environments often suffer from visual realism gaps and are constrained by limited asset diversity. As a result, policies trained on simulated data frequently underperform when deployed in the real world.

Recently, diffusion models (Wan et al., 2025; Kong et al., 2025; NVIDIA et al., 2025a; Yang et al., 2025; Zheng et al., 2024) have emerged as a promising method for generating realistic and diverse visual video. Several works have explored using diffusion models to generate vision-action data for policy training. Cosmos-Transfer1 (NVIDIA et al., 2025b) generates videos conditioned on semantic segmentation and depth, improving realism for sim-to-real transfer. RoboEngine (Yuan et al., 2025a) provides a flexible toolkit for generating diverse robot interaction scenes by combining background generation with accurate robot segmentation, without requiring camera calibration. RoboTransfer (Liu et al., 2025) further improves multi-view consistency by explicitly modeling 3D geometry using depth maps and surface normals, allowing controllable edits.

Despite these advances, two key challenges remain. First, most methods (NVIDIA et al., 2025b; Yuan et al., 2025a) generate videos from a single view, without ensuring consistency across viewpoints. This limits their usefulness for downstream robot tasks that rely on multi-camera inputs. RoboTransfer takes a step toward multi-view consistency, but its diversity is limited because it often transfers poorly to new domains. Second, existing works treat generated data as a static augmentation, without considering how to use it effectively during training.

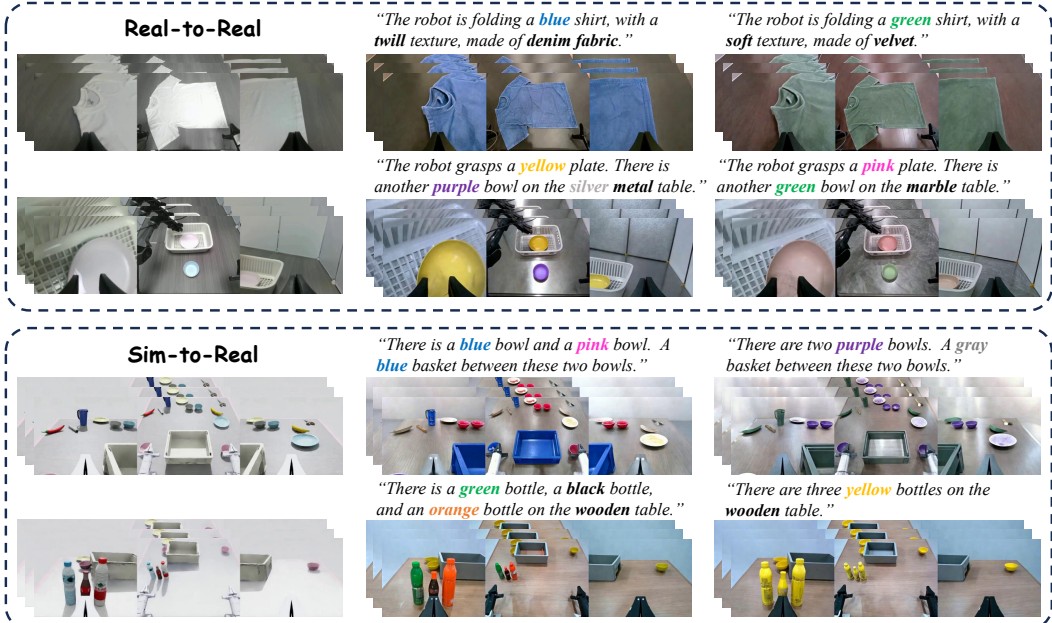

Figure 1: *DreamTransfer* demonstrates strong controllability in embodied manipulation video generation. It excels in text-controlled appearance editing while preserving 3D structure and geometric plausibility, and supports both real-to-real and sim-to-real transfer. The complete prompts used for generation is provided in the supplementary materials.

In this work, we propose **E**mbodied **M**anipulation **M**edia **A**daptation (*EMMA*), a VLA policy enhancement framework that integrates two core components: *DreamTransfer* and *AdaMix*. *DreamTransfer* is a diffusion Transformer (DiT)-based framework for generating multi-view consistent, geometrically grounded embodied manipulation videos. It jointly models appearance and geometry across multiple camera views, ensuring spatial and temporal coherence. *DreamTransfer* supports text-controlled visual transfer: users can edit the foreground objects, background, and lighting conditions of real or simulated demonstrations through natural language, while preserving the underlying 3D structure and geometrical plausibility of the scene. As illustrated in Figure 1, *DreamTransfer* enables realistic and controllable video generation for both real-to-real and sim-to-real transfer scenarios, making it a powerful tool for scalable robotic policy training. To improve policy learning, we further propose *AdaMix*, a hard-sample-aware training strategy. We define a set of functions to evaluate the quality of predicted trajectories from the VLA policy and use the performance score to drive an adaptive sampling mechanism. By iteratively refining the training distribution toward challenging cases, our method improves robustness and generalization.

We evaluate on a variety of robotic manipulation tasks in both video generation quality and real-world robot deployment, including `Fold Cloth`, `Clean Desk`, and `Throw Bottle`. These tasks span a wide range of challenges involving both rigid and deformable objects, short-horizon and long-horizon action sequences, and diverse skills such as grasping, pushing, placing, and draping. Compared to the state-of-the-art transfer model, *DreamTransfer* improves multi-view consistency by 42% and depth consistency by 24%, demonstrating superior geometric fidelity and cross-view coherence. In real-world robotic manipulation tasks involving zero-shot visual appearances, our method achieves over a 200% relative improvement in task success rate compared to training on real data alone, with an additional 13% gain when integrated with *AdaMix*.

In summary, our contributions are:

- We propose *EMMA*, a VLA policy enhancement framework that integrates a generative data engine with an effective training strategy. The data engine generates diverse, multi-view consistent robot manipulation videos for both rigid and deformable objects, while adaptive sample weighting improves VLA policy generalization.

- We propose *DreamTransfer*, a DiT-based model that generates multi-view consistent, geometrically grounded manipulation videos and supports text-controlled editing of foreground, background, and lighting conditions. We further introduce *AdaMix*, a hard-sample-aware training strategy that identifies challenging trajectories and adaptively reweights them during training.

- *EMMA* demonstrates strong performance in video generation and real-world robotic deployment. Compared to the state-of-the-art model, *DreamTransfer* achieves a 42% gain in multi-view consistency and a 24% gain in depth consistency, measured relatively. In zero-shot visual settings, our method achieves over a 200% performance gain compared to real-data training, with *AdaMix* providing an additional 13% improvement and enhancing cross-domain visual generalization.

## 2 METHOD

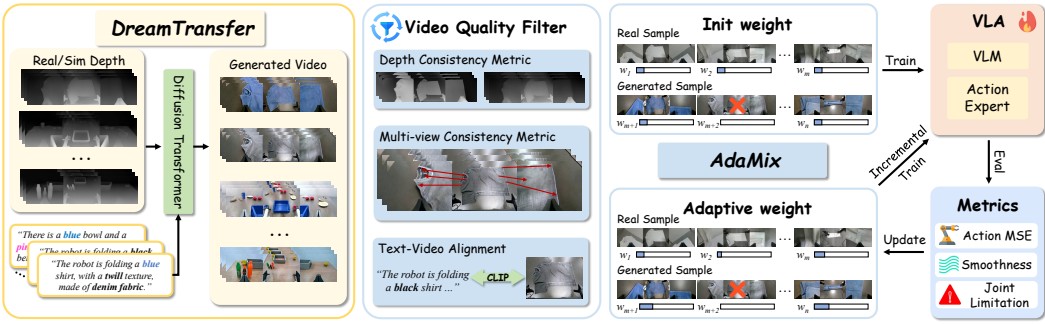

Figure 2: Overview of the *EMMA* framework. First, *DreamTransfer* generates multi-view consistent videos by performing text-controlled visual editing of the foreground, background, and lighting conditions, conditioned on depth and corresponding text prompts. The generated videos are then evaluated by a video quality filter. Low-quality videos are initially assigned zero sampling weight to stabilize early-stage training. The *AdaMix* module further adaptively reweights training samples based on trajectory performance metrics, up-weighting challenging samples to improve policy robustness and generalization.

### 2.1 FRAMEWORK OVERVIEW

We propose *EMMA*, a VLA policy enhancement framework that integrates two core components: *DreamTransfer* and *AdaMix*. As illustrated in Figure 2, *DreamTransfer* functions as a generative data engine for training VLA, capable of generating multi-view consistent, geometrically grounded robot manipulation videos from both real and simulated inputs. The framework supports fine-grained text-controlled editing of visual attributes such as foreground objects, background scenes, and lighting conditions, while preserving the underlying 3D geometry and action dynamics.

The pipeline begins with real or simulated demonstration videos, which are processed by *DreamTransfer* that generates robot manipulation videos across multiple camera views. The model ensures spatial and temporal coherence by jointly modeling multi-view appearance and geometry, which also effectively bridging the visual gaps between simulation and reality. The generated videos are subsequently filtered according to depth consistency, multi-view consistency, and text-video similarity metrics. To ensure training stability, we initialize the training process using only high-quality videos by setting the sampling probability of low-quality videos to zero. To further enhance policy generalization, we introduce *AdaMix*, a hard-sample-aware training strategy that dynamically adjusts the sampling weights based on their trajectory performance metric. By adaptively emphasizing challenging cases, *AdaMix* enhances policy generalization and improves performance on real-world

environments. Our framework advances effective policy learning for embodied manipulation by combining high-fidelity video generation with the adaptive sample selection of *AdaMix*.

We first introduce the *DreamTransfer* model in Section 2.2, followed by a presentation of the *AdaMix* training strategy in Section 2.3.

## 2.2 DREAMTRANSFER

The overall framework of *DreamTransfer* is illustrated in Figure 3. *DreamTransfer* features a dual-branch architecture, both built upon DiT-based framework (NVIDIA et al., 2025a;b). The main branch progressively denoises latent video tokens, while the parallel ControlNet (Zhang et al., 2023) branch enhances geometric consistency by incorporating depth-based structural guidance.

*DreamTransfer* leverages the multi-view in-context learning capabilities (Jang et al., 2025; Huang et al., 2024; Zhao et al., 2024; Liu et al., 2025) inherently present in pre-trained diffusion models, which ensures spatial and temporal coherence across different camera viewpoints. First, multiple synchronized video depth from different viewpoints $\{v^{(1)}, v^{(2)}, \ldots, v^{(M)}\}$ are concatenated along the width dimension and encoded into a unified latent representation using a VAE encoder (Kingma & Welling, 2022):

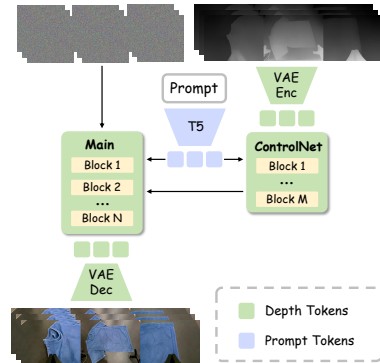

Figure 3: Overview of the *Dream-Transfer* framework. Multi-view depth maps are concatenated along the width dimension. The main branch denoises latent video tokens, while a parallel ControlNet branch ensures geometric consistency by incorporating depth constraints.

$$d_{init} = \text{Enc}([v^{(1)}, v^{(2)}, \ldots, v^{(M)}]).$$

In parallel, a pretrained T5 text encoder (Raffel et al., 2023) converts text prompt into high-dimensional semantic embeddings. These textual features are fused with the visual latents via cross-attention, enabling fine-grained control over object appearance. The main branch then takes the noisy latent $z_t$ at time step $t$ together with the prompt features $s$ and depth features $d_t$, to predict the denoised latent:

$$n = f_\theta(z_t, t, d_t, s),$$

where $f_\theta$ denotes the diffusion transformer parameterized by $\theta$, and $n$ represents the predicted denoised latent. Finally, the predicted denoised latent is decoded by the VAE decoder to reconstruct the output video, which both preserves the underlying 3D structure and faithfully reflects the prompt-specified appearance changes.

## 2.3 ADAMIX

Despite extensive fine-tuning on a large mixture of real-world demonstrations and generated data, VLA model still exhibits errors when evaluated on the training set. This observation reveals that uniform training paradigms fail to adequately address challenging, long-tail scenarios, even when those scenarios are present in the training data. To bridge this gap, we collect challenging sample from model evaluations on the training set.

Building on this insight, we present *AdaMix*, a hard-sample-aware training strategy that dynamically reweights training data based on policy performance. Unlike uniform sampling schemes that treat all samples equally, *AdaMix* continuously identifies challenging scenarios, particularly those in the long tail where policies typically struggle (Ma et al., 2024). By up-weighting these hard samples during training, *AdaMix* achieves improved generalization.

The *AdaMix* pipeline consists of three stages, as illustrated in right part of Figure 2. First, generated videos are evaluated by a video quality filter based on depth consistency, multi-view consistency, and text-video similarity. Samples failing to meet quality thresholds are assigned zero sampling weight, ensuring that only geometrically grounded and semantically faithful data are used for training. Then, the VLA model is initially trained using uniform sampling over real demonstrations and high-quality generated videos. After the loss converges, we compute three carefully designed metrics on each training sample to identify challenging instances:

**Action Prediction Error:** We compute the Mean Squared Error (MSE) between the predicted action chunk $\hat{a}_{i:t}$ and the corresponding ground-truth sequence $a_{i:t}$ over a window of at most $L$ frames:

$$r_i^{\text{MSE}} = -\frac{1}{L} \sum_{t=0}^{L-1} \|\hat{a}_{i+t} - a_{i+t}\|_2^2.$$

**Trajectory Smoothness:** To encourage physically plausible actions, we measure the second-order difference of joint angles to penalize abrupt joint actions:

$$r_i^{\text{Smooth}} = -\sum_j \left| \frac{a_{i+2,j} - 2a_{i+1,j} + a_{i,j}}{(\Delta t)^2} \right|,$$

where $j$ indexes the joint dimension, and $\Delta t$ represents the time interval between frames.

**Joint Angle Limitation:** We assign a binary indicator to ensure safety, setting

$$r_i^{\text{Limit}} = \begin{cases} 1, & \text{if all joint angles within thresholds,} \\ 0, & \text{otherwise.} \end{cases}$$

These scores are min-max normalized to $\tilde{r}_i^{(\cdot)} \in [0,1]$ and combined into a unified score per sample :

$$s_i = \frac{\tilde{r}_i^{\text{MSE}} + \tilde{r}_i^{\text{Smooth}} + \tilde{r}_i^{\text{Limit}}}{3},$$

Empirically, we find that a balanced combination is sufficient to identify challenging and informative training samples without the need for manual tuning or learned weights. A higher $s_i$ indicates better overall performance on the $i$-th sample.

During incremental training, sampling weights are updated as:

$$p(i) \propto \gamma + \lambda \cdot (1 - s_i),$$

where $\gamma > 0$ ensures minimum support for all samples, and $\lambda$ controls the emphasis on hard samples.

By adaptively up-weighting samples where the policy performs poorly, the training distribution gradually shifts toward underperforming regions while preserving data diversity. Note that all evaluations and re-weighting are performed strictly on the training set to prevent any potential leakage from validation data.

## 3 EXPERIMENTS

In this section, we evaluate *EMMA* framework on both video generation and real-world robotic deployment. We conducted extensive experiments on diverse tasks such as `Fold Cloth`, `Clean Desk`, and `Throw Bottle`. The first task is real-to-real transfer, and the last two tasks are sim-to-real. These tasks cover a range of robotic tasks involving both rigid and deformable objects, long-horizon and short-horizon actions, and different manipulation skills such as grasping, pushing, and placing.

We first evaluate the quality of generated videos from *DreamTransfer* in Section 3.1. Then we use the generated data to train downstream VLA policy model and deploy on real-world robot in Section 3.2.

### 3.1 VIDEO GENERATION QUALITY

**Implementation Details.** We evaluate our method against two state-of-the-art models for robot manipulation video transfer: Cosmos-Transfer1 (NVIDIA et al., 2025b) and RoboTransfer (Liu et al., 2025). Both *DreamTransfer* and RoboTransfer natively support multi-view transfer, while Cosmos-Transfer1 is only designed for single-view generation. To ensure a fair comparison across all three models, we adopted model-adapted input and output processing strategies: for Cosmos-Transfer1, we processed each camera view independently and then concatenated the generated frames along the width dimension to form multi-view outputs; for*DreamTransfer* and RoboTransfer, we directly

Table 1: Comparison of video generation quality on robot manipulation tasks. The metrics demonstrate that *DreamTransfer* outperforms others in terms of multi-view consistency, depth consistency and text-video alignment. Best results are in **bold**, second-best are in underlined.

| Task | Model | Pix.Mat.($\uparrow$) | RMSE($\downarrow$) | Abs.Rel.($\downarrow$) | Sq.Rel.($\downarrow$) | CLIPSim.($\uparrow$) |
|------|-------|------------|---------|-----------|----------|-----------|
| Fold Cloth | RoboTransfer | 1736 | 3.97 | 0.50 | 2.13 | 24.63 |
| | Cosmos-Transfer1 | 2210 | 2.78 | 0.36 | 1.14 | 25.02 |
| | *DreamTransfer* | 2604 | 2.50 | 0.32 | 0.97 | 24.54 |
| Clean Desk | RoboTransfer | 3213 | 2.50 | 0.32 | 0.85 | 23.87 |
| | Cosmos-Transfer1 | 2484 | 1.55 | 0.19 | 0.36 | 25.02 |
| | *DreamTransfer* | 4311 | 1.45 | 0.18 | 0.33 | 25.75 |
| Throw Bottle | RoboTransfer | 1944 | 2.08 | 0.34 | 0.93 | 23.31 |
| | Cosmos-Transfer1 | 1597 | 1.71 | 0.26 | 0.64 | 23.79 |
| | *DreamTransfer* | 2894 | 1.36 | 0.20 | 0.33 | 23.74 |
| Average | RoboTransfer | 2298 | 2.85 | 0.39 | 1.30 | 23.94 |
| | Cosmos-Transfer1 | 2097 | 2.01 | 0.27 | 0.71 | 24.61 |
| | *DreamTransfer* | **3270** | **1.77** | **0.23** | **0.54** | **24.68** |

used the concatenated multi-view videos as input, enabling them to generate multi-view videos in a single inference pass.

Our evaluation focuses on three key aspects: multi-view consistency, depth consistency, and text-to-video alignment. Multi-view consistency measures the geometric coherence of generated scenes across different camera angles. Depth consistency evaluates the fidelity of predicted depth against ground truth. Text-to-video alignment assesses how well the generated video matches the input task instruction. Input conditions are kept identical across methods to ensure a controlled comparison. More details are provided in the Appendix A.4

**Results analysis.** As shown in Table 1, *DreamTransfer* achieves the best performance across all metrics in most tasks and consistently outperforms both RoboTransfer and Cosmos-Transfer1 in multi-view consistency and geometric fidelity. On average, *DreamTransfer* improves pixel matching by 42% over the second-best method (RoboTransfer). This demonstrates that our multi-view conditioned generation produces highly consistent appearances across camera views. In depth consistency, *DreamTransfer* reduces Squared Relative Error to 0.54, a 24% improvement over Cosmos-Transfer1 and 38% over RoboTransfer. Similarly, it achieves the lowest relative errors, indicating more accurate and geometrically grounded 3D structures. These gains are enabled by our explicit depth conditioning, which enforces cross-view structural coherence during video generation. Regarding text-video alignment, *DreamTransfer* scores 24.68 on average, surpassing both baselines and showing no loss in text prompt semantic alignment. Viusal comparisons of the generated videos are provided in Appendix A.1.

## 3.2 REAL-WORLD ROBOT EVALUATION

We conduct a series of experiments to evaluate our *EMMA* framework on real-world robotic manipulation tasks. We focus on three key questions: (1) Can co-training with generated data improve real-world policy performance and generalization to novel object appearances and environments? (2) How does the mixing ratio of real and generated data affect policy performance? (3) Can *AdaMix*, our hard-sample-aware adaptive training strategy, further enhance real-world policy performance?

**Implementation Details.** We adopt $\pi_0$ (Black et al., 2024) as our base VLA policy model architecture without modifications and perform post-training on the pre-trained model. We evaluate our framework on three challenging real-world robotic tasks: `Fold Cloth`, `Clean Desk`, and `Throw Bottle`. We train policies on a mixed dataset $D^\alpha$, composed of real data $D_R$ and generated data $D_G$. During training, each sample is drawn from $D_G$ with probability $\alpha$ and from $D_R$ otherwise, where $\alpha$ serves as the data mixing ratio (Wei et al., 2025). To ensure a fair comparison, all experiments for a given task are trained under the same configuration, including batch size, learning rate, data composition, and training steps. Traditional robot policy learning evaluation focuses on task success rate, but this binary indicator often cannot fully reflect the performance of the policy.

Table 2: Effect of video generation models on downstream VLA policy performance in real-world robot tasks. For video generation models, policies are trained with a 50% mixing ratio of real and generated data. No.Aug. denotes training on real data only, without any augmentation. Score is the behavior score; SR is success rate. Result is averaged over 5 trials across 4 distinct visual variations of the foreground object, totaling 20 runs per task. Best results are in **bold**.

| Model | Fold Cloth | | Clean Desk | | Throw Bottle | | Average | |
|---|---|---|---|---|---|---|---|---|
| | Score | SR | Score | SR | Score | SR | Score | SR |
| No.Aug. | 3.0 | 10% | 4.3 | 65% | 2.0 | 10% | 3.1 | 28% |
| Cosmos-Transfer1 | 3.3 | 40% | 4.1 | 70% | 3.2 | 40% | 3.5 | 50% |
| *DreamerTransfer* | **4.4** | **65%** | **4.6** | **80%** | **3.3** | **50%** | **4.1** | **65%** |

We use a dual evaluation metric of behavior score and success rate to avoid masking performance differences with a single sparse success rate. Each reported result is averaged over 5 trials and 4 distinct visual variations of the foreground object, resulting in 20 evaluation runs per setting. More details of real-world robot experiments can be found in Appendix A.5.

**Experimental Platform.** Experiments run on an Agilex CobotMagic platform with two PiPER arms and three Intel RealSense D435i cameras (two wrist-mounted, one head-mounted).

### 3.2.1 COMPARISON WITH BASELINE GENERATION MODELS AND GENERALIZATION TESTS

As shown in Table 2, the choice of video generation model significantly impacts the performance of downstream VLA policy in real-world robotic tasks. Training without data augmentation yields the lowest performance across all tasks, highlighting the necessity of generated data for policy generalization under novel visual conditions.

When using generated data for training, there are clear performance differences depending on the quality and realism of the generated videos. Policies trained with data augmented by Cosmos-Transfer1 show moderate improvements over the no-augmentation baseline. This is particularly evident in success rate, with gains of 22%, but performance still falls short in handling complex deformable object manipulation, as shown by the relatively low success rate (40%) on `Fold Cloth`. In contrast, *DreamerTransfer* achieves consistent and substantial improvements across all three tasks, outperforming both the baseline and Cosmos-Transfer1 in both behavior score and success rate. Notably, on `Fold Cloth`, *DreamerTransfer* achieves a 65% success rate, more than doubling that of the baseline without augmentation. This demonstrates that multi-view consistent, geometrically consistent video generation enables more effective policy learning, especially for challenging tasks involving non-rigid dynamics.

### 3.2.2 IMPACT OF GENERATED DATA MIXING RATIO ON REAL-WORLD ROBOT PERFORMANCE

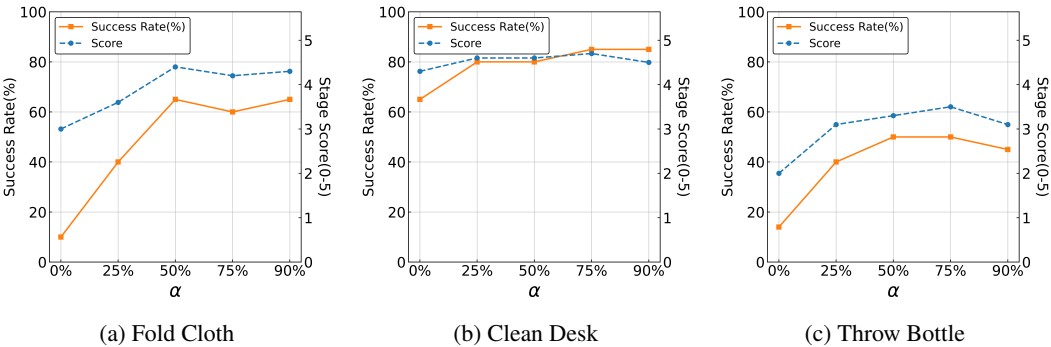

| (a) Fold Cloth | (b) Clean Desk | (c) Throw Bottle |

Figure 4: Impact of data mix ratios on real-world robotic tasks performance.

Table 3: Real-world performance comparison between the fixed mixing ratio sampling ($\alpha = 50\%$, FixMix) and *AdaMix* with adaptive weight sampling. In the FixMix baseline, sampling weights remain constant throughout training. Results show that *AdaMix* achieves consistent improvements in both behavior score and success rate across all tasks and on average. Best results are in **bold**.

| Method | Fold Cloth | | Clean Desk | | Throw Bottle | | Average | |
|--------|------------|----|------------|----|--------------|----|---------|----|
| | Score | SR | Score | SR | Score | SR | Score | SR |
| FixMix | 4.4 | 65% | 4.6 | 80% | 3.3 | 50% | 4.1 | 65% |
| *AdaMix* | **4.6** | **75%** | **4.8** | **90%** | **4.4** | **70%** | **4.6** | **78%** |

Table 4: Comparison of execution time (Time, seconds), trajectory smoothness (Smth., angular acceleration in $^\circ/s^2$) and joint overlimit (JOL., frames) between FixMix and *AdaMix* training strategies on real-world robotic tasks. Lower values are better for all metrics. Best results are in **bold**.

| Method | Fold Cloth | | | Clean Desk | | | Throw Bottle | | | Average | | |
|--------|------|-------|------|------|-------|------|------|-------|------|------|-------|------|
| | Time | Smth. | JOL. | Time | Smth. | JOL. | Time | Smth. | JOL. | Time | Smth. | JOL. |
| FixMix | 40.4 | 2.3 | 57 | 12.7 | 2.6 | 66 | 46.0 | 1.3 | 15 | 33.0 | 2.0 | 46 |
| *AdaMix* | **39.8** | **2.2** | **54** | **11.3** | **2.4** | **53** | **38.9** | **1.2** | **10** | **30.0** | **1.9** | **39** |

We study how the mixing ratio between real and generated data affects policy performance, while keeping the total amount of training data and the number of training steps fixed across all mixing ratios for a given task. Results in Figure 4 reveal two key insights.

As shown in Figure 4, performance improves significantly when generated data is introduced, peaking at a 50% mixing ratio. This balanced mix achieves optimal generalization, particularly on appearance-sensitive tasks like `Fold Cloth`, where success rate jumps from 10% (0% generated data) to 65%. Beyond 50%, performance plateaus: increasing the proportion of generated data to 75% or 90% yields no further gain in average success rate and slightly reduces consistency. Notably, at 90% generated data, the `Throw Bottle` task exhibits a drop in success rate (from 50% to 45%), suggesting that excessive reliance on generated data may propagate subtle visual or dynamic inaccuracies that harm performance in fast, precision-critical tasks.

### 3.2.3 ABLATION STUDY ON ADAMIX TRAINING FRAMEWORK

**Implementation Details.** To evaluate the effectiveness of the *AdaMix* training strategy in improving real-world policy performance, we conduct a controlled incremental training experiment. Both methods are trained for the same number of steps on each task, using an identical dataset comprising real-world demonstrations and generated videos. The training process begins with the same initialization: all samples that pass the video quality filter are assigned equal initial weights, while low-quality samples are assigned zero weight and excluded from training. The key difference lies in how sampling weights evolve during training. In the FixMix baseline, sampling weights remain constant throughout training, implementing uniform sampling over the retained data. In *AdaMix*, after half of the training steps, sampling weights are dynamically updated based on trajectory performance scores, up-weighting challenging and informative samples.

Our results, summarized in Table 3, demonstrate the effectiveness of adaptive sampling: *AdaMix* achieves a 13% improvement in average success rate. Beyond task completion, the policy trained with *AdaMix* exhibits superior low-level control, as quantified in Table 4. On average, it completes tasks 3.0 seconds faster, reduces joint limit violations by 7.0 counts, and produces smoother trajectories with a 0.1 lower smoothness. These gains across all tasks, especially in long-tail scenarios such as `Fold Cloth` and `Throw Bottle`, indicate that the policy-performance-based metrics effectively identify challenging samples where the policy struggles. By dynamically reweighting these samples, *AdaMix* enables more targeted learning. The key insight is that not all data are equally informative for refinement, and uniform sampling underutilizes the potential of challenging cases. By adaptively focusing on hard samples, our method implicitly constructs a curriculum that aligns with the policy's current weaknesses, enhancing both task performance and execution quality.

This demonstrates that hard-sample-aware training is a powerful mechanism for real-world policy improvement. Visual evidence of real-world deployment is provided in Appendix A.2.

## 4 RELATED WORK

### 4.1 VISUAL GENERALIZABLE IMITATION LEARNING

Imitation learning enables visuomotor policies to learn from human demonstrations, providing an effective pathway for robotic manipulation (Chi et al., 2024; Li et al., 2025b; Jiang et al., 2025). Recently, VLA models have significantly improved generalization by integrating semantic understanding into action generation (Brohan et al., 2023; Kim et al., 2024). Early systems like CLIPort (Shridhar et al., 2021) established the foundation for vision-conditioned control, while subsequent works enhanced capabilities through chain-of-thought reasoning (Zhen et al., 2025). Recent advances include domain specialization (Yue et al., 2024), occlusion handling (Wei et al., 2024), and safety-aware execution (Zhang et al., 2025). Models like OpenVLA and EF-VLA (Huang et al., 2025) further improve performance through dual visual encoders or preserved semantic alignment, while self-correcting frameworks (Li et al., 2025a) enable recovery from failures in cluttered environments. However, VLA models require large-scale, diverse training data to generalize across objects and environments, and suffer from poor out-of-distribution performance when trained on limited real-world demonstrations.

Collecting data across diverse objects and environments is expensive and time-intensive (Collaboration et al., 2025; Khazatsky et al., 2025). This gap motivates the use of generated data as a scalable means to enrich visual diversity, provided that generated content preserves geometrical plausibility and spatial coherence.

### 4.2 GENERATIVE MODELS FOR EMBODIED DATA SYNTHESIS

To improve generalization in VLA models, various generative approaches have been proposed to generate diverse robot data at low cost (Lin et al., 2025; Chen et al., 2024a; Jin et al., 2025; Yuan et al., 2025b; Wang et al., 2025). While traditional data augmentation techniques remain effective for in-domain generalization (Chi et al., 2024), they often struggle under significant visual distribution shifts. In contrast, generative models offer stronger cross-domain adaptation potential (Teoh et al., 2024), yet many methods rely on additional inputs such as object masks or scene-specific annotations (Chen et al., 2024b; Mandi et al., 2023; Wang et al., 2024). Furthermore, techniques based on inpainting or scene completion can exhibit instability across varied environments, frequently requiring per-scene hyperparameter tuning to ensure reliable generation (Zhuang et al., 2024; Yu et al., 2023). Recent advances in diffusion models (Ho et al., 2020; Song et al., 2022; Ho & Salimans, 2022; Blattmann et al., 2023) and video diffusion transformers (Lu et al., 2023; Yang et al., 2025) have enabled high-fidelity generation of embodied interaction sequences. Cosmos-Transfer1 (NVIDIA et al., 2025b) generates realistic scenes conditioned on semantic segmentation and depth maps, effectively narrowing the sim-to-real gap through structured visual cues. Robo-Dreamer (Zhou et al., 2024) supports text-guided future trajectory generation, enabling language-conditioned behavior generation, though sometimes at the cost of geometric fidelity. Similarly, RoboTransfer (Liu et al., 2025) incorporates depth and surface normal predictions to enhance cross-view alignment. However, its dependence on a fixed training distribution constrains visual diversity and impedes generalization when deployed in novel environments.

## 5 CONCLUSION

In this paper, we address the challenge of scaling up diverse and generalizable VLA learning for robot manipulation, where real-world data collection is costly and simulation lacks visual realism. We present *EMMA*, a framework for enhancing VLA policy via text-controlled embodied manipulation video generation with *DreamTransfer* and adaptive training with *AdaMix*. We evaluate *EMMA* on real-world robotic tasks with rigid and deformable objects under diverse visual conditions. *EMMA* achieves over a 200% improvement in task success rate compared to training on real data alone, with an additional 13% gain from *AdaMix*. These results demonstrate that *EMMA* provides an effective pathway to enhancing the generalization of VLA policies.

## 6 REPRODUCIBILITY STATEMENT

We are committed to ensuring the reproducibility of our results. The proposed *DreamTransfer* and *AdaMix* modules are described in detail in Section 2.2 and Section 2.3, respectively. The data construction pipeline for *DreamTransfer* is detailed in Appendix A.3.1, and its full training configuration, including hyperparameters and optimization settings, is provided in Appendix A.3.2. Evaluation metrics for generated videos are specified in Appendix A.4. For the downstream real-world robotic experiments, task descriptions for real-world deployments are given in Appendix A.5.1, and the behavior score criteria used for evaluation are defined in Appendix A.5.3. The VLA model training configuration is fully listed in Appendix A.5.2. To support replication and future research, we will release our code.

## 7 STATEMENT ON THE USE OF LARGE LANGUAGE MODELS

During the preparation of this work, the authors made use of large language models (LLMs), including ChatGPT (OpenAI) and Qwen (Alibaba Cloud), as auxiliary tools to assist with language refinement. These models were employed solely for improving the clarity, grammar, and stylistic coherence of the manuscript. No part of the scientific reasoning, method design, data interpretation, or technical content was generated or influenced by the models. All text has been thoroughly reviewed, revised, and approved by the authors, who bear full responsibility for the accuracy and integrity of the final content.

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

# A APPENDIX

## A.1 VISUALIZATION RESULTS COMPARED TO STATE-OF-THE-ART ROBOT MANIPULATION VIDEO GENERATION MODELS

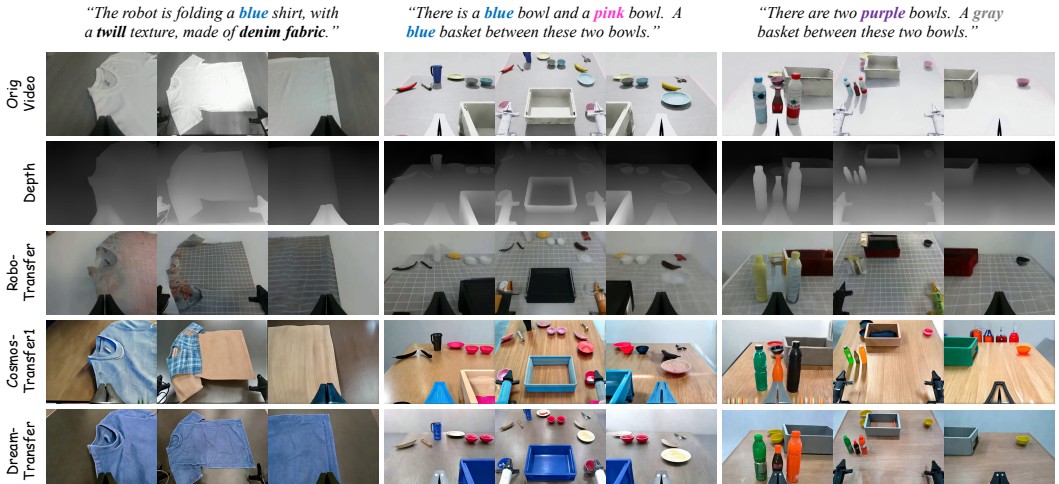

Figure 5: Visualization results compared to the state-of-the-art robot manipulation video transfer models. The results demonstrate that *DreamTransfer* significantly outperforms other models. *DreamTransfer* generates videos with superior multi-view consistency, more accurate 3D structure preservation, and higher geometrical plausibility under text-controlled appearance editing.

## A.2 DEMONSTRATION OF REAL-WORLD POLICY DEPLOYMENT ON ROBOT

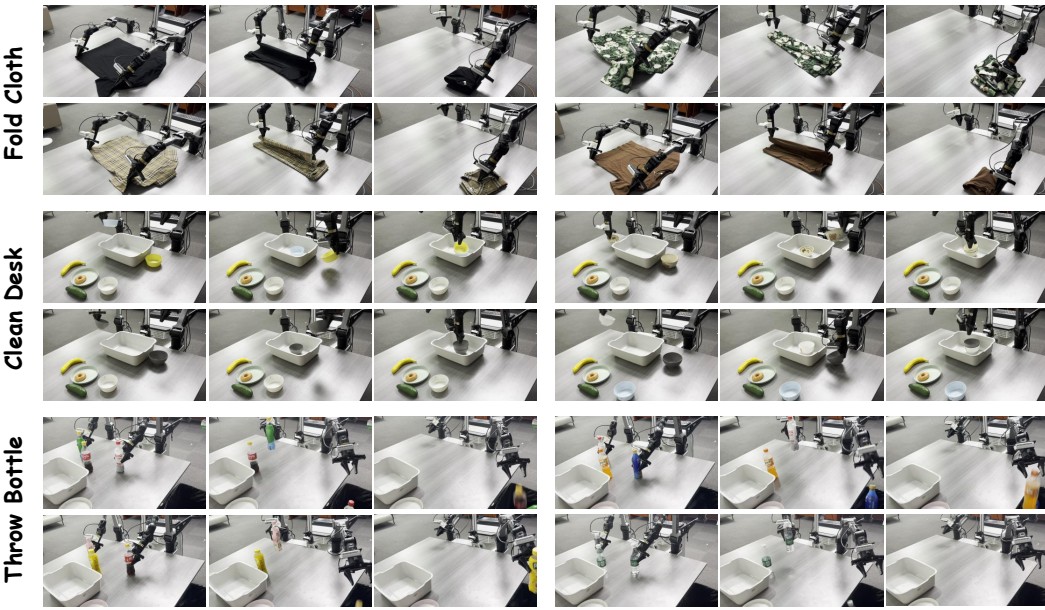

Figure 6: Real-world deployment of *AdaMix*-trained policies on `Fold Cloth`, `Clean Desk`, and `Throw Bottle`. The robot exhibits strong generalization across appearance variations. See supplementary materials for full videos.

### A.3 The details of the DreamTransfer

#### A.3.1 Dataset Construction

To enable multi-view modeling, we construct a dataset of 50k generated multi-view video clips based on the Agibot World dataset (Bu et al., 2025), covering 36 diverse scenes. Each clip contains aligned multi-view RGB frames, temporally consistent depth maps, and a text caption. Depth maps are generated using the state-of-the-art estimator Video Depth Anything (Chen et al., 2025), ensuring geometric coherence across time. To support fine-grained appearance control, we design caption templates that explicitly describe foreground, background, and lighting conditions. These templates are automatically filled by Qwen2.5-VL-7B-Instruct (Bai et al., 2025), yielding high-quality, appearance-focused captions.

#### A.3.2 Training Details

We fine-tune the pretrained Cosmos-Transfer1 model (NVIDIA et al., 2025b) on our multi-view robotic manipulation dataset. The real demonstrations are collected on the AgileX CobotMagic platform equipped with two PiPER arms and three Intel RealSense D435i cameras. The simulated demonstrations are collected in the NVIDIA Isaac Sim environment. Each camera captures RGB images at a resolution of $640 \times 480$, and during training, we concatenate the three views along the width dimension to form a single input frame of size $1920 \times 480$.

However, this concatenated resolution exceeds the maximum input size supported by the Cosmos-Transfer1 pretrained model, leading to visible artifacts and noise in the generated video. To address this, we adopt a two-stage fine-tuning strategy designed to: (1) stabilize learning of multi-view geometric and appearance consistency at a reduced resolution, and (2) progressively adapt the model to full-resolution inputs while recovering high-fidelity details in the out-of-distribution view. The detailed training configurations for each stage are summarized in 5.

Table 5: Training configurations for the two-stage fine-tuning process.

| Configurations | Stage 1 | Stage 2 |
|---|---|---|
| Input Resolution (W $\times$ H) | $576 \times 128$ | $1920 \times 480$ |
| Batch Size | 32 | 4 |
| Training Steps | 3500 | 4500 |
| Optimizer | AdamW | AdamW |
| Learning Rate | $1 \times 10^{-5}$ | $1 \times 10^{-5}$ |
| Trainable Parameters | Main Branch + ControlNet | Main Branch + ControlNet |
| Purpose | Multi-view consistency | High-res adaptation |

### A.4 Video Generation Quality Evaluation and Filtering

To ensure the realism and fidelity of generated videos for downstream policy training, we evaluate and filter them using three key metrics: multi-view consistency, depth consistency, and text-video alignment. These metrics assess geometric plausibility, spatial coherence, and semantic faithfulness to the input prompt, respectively. They are used both for quantitative evaluation (Section 3.1) and as criteria for filtering low-quality generations before policy training.

**Multi-View Consistency.** We measure geometric and appearance coherence across camera views using a state-of-the-art image matcher (Shen et al., 2024). For each video, we compute the number of matching pixels (Mat.Pix.) between the center view and left/right views across frames. Higher match counts indicate better view consistency and are used to filter out videos with misaligned or distorted geometry.

**Depth Consistency.** To evaluate 3D structural plausibility, we extract temporally coherent depth maps from both the original and generated videos using Video Depth Anything (Chen et al., 2025). We compute scale-invariant metrics including Root Mean Squared Error (RMSE), Absolute Relative Error (Abs.Rel.), and Squared Relative Error (Sq.Rel.).

**Text-Video Alignment.** We assess semantic fidelity using CLIP (Radford et al., 2021) similarity. The input prompt is decomposed into three components: foreground object, background scene, and lighting condition. For each frame and view, we compute CLIP similarity between the image and each prompt component. Scores are averaged across views and time steps. Higher values indicate better alignment with the intended semantics, and only videos above a similarity threshold are kept for training.

## A.5 THE DETAILS OF REAL-WORLD ROBOT EXPERIMENTS

### A.5.1 ROBOT TASK DESCRIPTION

We conducted experiments on the following three real robot tasks, covering real-to-real and sim-to-real.

For the real-to-real setting, we use the `Fold Cloth` task. This is a long-horizon, multi-stage task that involves manipulating a deformable object. The task has two main phases: folding and pushing. Initially, a piece of cloth is placed flat on a table. Two robot arms must cooperate to fold the cloth. After folding, the right arm pushes the cloth to a target location on the table.

For the sim-to-real setting, we evaluate on two simulated tasks: `Clean Desk` and `Throw Bottle`. In `Clean Desk`, there is a box and two bowls on a table. The two robot arms must work together to place the bowls into the box. In `Throw Bottle`, three bottles are placed on the table, and the arms must pick and throw them one by one into a trash bin beside the table.

We collect 50 real-world demonstration data for `Fold Cloth` and 20 for each of the other two tasks. For `Fold Cloth`, we apply video transfer method to the real-world demonstration videos, generating 50 corresponding generated data. To evaluate sim-to-real transfer, we additionally collect 20 demonstration trajectories for `Clean Desk` and `Throw Bottle` in the NVIDIA Isaac Sim environment, which are then transferred to match the target real-world domain, yielding 20 photorealistic generated data per task.

### A.5.2 TRAINING DETAILS OF THE VLA MODEL

Table 6: Training configurations for the $\pi_0$ policy model on three robotic manipulation tasks.

| Configuration | Fold Cloth | Clean Desk | Throw Bottle |
|---|---|---|---|
| Batch Size | 64 | 64 | 64 |
| Training Steps | 20000 | 5000 | 10000 |
| Optimizer | AdamW | AdamW | AdamW |
| Warmup Steps | 1000 | 1000 | 1000 |
| Init Learning Rate | $2.5 \times 10^{-8}$ | $2.5 \times 10^{-8}$ | $2.5 \times 10^{-8}$ |
| Learning Rate Schedule | Cosine Decay | Cosine Decay | Cosine Decay |
| Trainable Parameters | All | All | All |

### A.5.3 BEHAVIOR SCORE EVALUATION CRITERIA

The behavior score is a fine-grained, instruction-aligned metric that measures task progress based on observable actions. It provides a more nuanced evaluation than binary success/failure by allowing partial credit. The maximum score for each task is 5. Final scores are computed as:

$$\text{Score} = 5 + \sum (\text{deductions})$$

The deduction rules for the behavior score are specified task-by-task as follows, with each rule clearly listing the corresponding failure scenario and its deduction value:

**1. Fold Cloth**
One corner not grasped: $-2$
Unable to move to target location: $-2$

Two or more corners not grasped: $-4$
No meaningful progress or no object interaction: $-5$

**2. Clean Desk**
One bowl not picked up: $-2$
Two bowls not picked up: $-4$
No meaningful progress or no object interaction: $-5$

**3. Throw Bottle**
One bottle not picked up: $-2$
Two bottles not picked up: $-4$
Three bottles not picked up: $-4$
No meaningful progress or no object interaction: $-5$

