# OpenReview forum: "EMMA: Generalizing Real-World Robot Manipulation via Generative Visual Transfer"
_ICLR.cc/2026/Conference — ICLR 2026 Conference Withdrawn Submission_

### Official Review · Reviewer_zySs · 2025-10-28

**Soundness:** 3
**Presentation:** 2
**Contribution:** 2
**Rating:** 4
**Confidence:** 4

**Summary:**

This paper proposes a generative model that produces a large number of videos based on existing ones by augmenting visual features such as color, texture, and materials. This approach improves the success rate of robotic tasks. The generated videos are depth- and multi-view-consistent, showing strong alignment with the accompanying text. The paper also introduces an adaptive sampling approach with sound metrics that filters out low-quality generated videos and places greater focus on high-error samples, further enhancing the performance of robotic tasks.

**Strengths:**

The video generative model surpasses the baselines in terms of consistency and semantic alignment, leading to improved performance in robotic tasks. The proposed adaptive sampling approach further enhances this performance.

**Weaknesses:**

1. The video generation fine-tuning strategy is quite common, and the sampling strategy for VLA fine-tuning lacks novelty.
2. The paper proposes a video quality filter and introduces three additional metrics in Adamix. How do these hyperparameters influence performance? Does this sampling strategy lead to fewer fine-tuning steps for the VLA?

**Questions:**

1. Table 4 reports the execution time. Does this execution time refer to how long the robot takes to complete the task? Why is the execution time influenced by the sampling approach? AdaMix results in shorter execution time — what causes this improvement?
2. How are the videos generated, and what prompts are used? Does the original video dataset cover the scene distributions, such as color and texture? Does the generated video dataset cover the scene distributions, such as color and texture?

---

### Official Review · Reviewer_Mpww · 2025-10-29

**Soundness:** 2
**Presentation:** 2
**Contribution:** 2
**Rating:** 2
**Confidence:** 3

**Summary:**

The paper introduces a framework designed to improve policy learning for real-world robotic manipulation by applying data augmentation to demonstrations. EMMA combines two main components. The first, DreamTransfer, is a diffusion-based generative model that produces multi-view, geometrically consistent, and text-controllable manipulation videos from real or simulated demonstrations, aiming to bridge the visual gap between simulation and reality. The second, AdaMix, is a hard-sample-aware adaptive training strategy that dynamically reweights training data based on policy performance metrics. The method is evaluated on three real-world manipulation tasks using a pre-trained Pi0 model. Experimental results show that incorporating DreamTransfer-generated data improves task success rates compared to training only on real data, and that AdaMix further enhances performance.

**Strengths:**

The proposed approaches for data augmentation and hard sample mining are sound.

Experiments demonstrate that data augmentation can improve policy performance.

Some minimal ablation studies are reported.

**Weaknesses:**

There is little to no novelty in this work. The proposed approach appears to simply be CosmosTransfer adapted for multi-view inference by concatenating images from multiple cameras along the width dimension. If there are other differences then they are not apparent from the method's description.

As mentioned above, the method section is missing lots of critical details and essentially only describes the high-level idea of the method. It is also does not described at all how the method is trained.

The proposed AdaMix is a relatively standard hard-sample mining/adaptive data reweighting strategy. The novelty lies mainly in engineering adaptation to robot policy training (choice of metrics, integration with DreamTransfer-generated data). Many important details are missing as well. The exact values or scheduling of lambda and minimum weight hyper-parameters are not detailed. The metrics (e.g., smoothness and limit violations) are hand-crafted heuristics — their sensitivity and weighting aren’t justified beyond “empirical balance”. Without ablation results on these choices, it’s difficult to assess whether AdaMix provides consistent improvements or just coincidental gains due to hyper-parameter tuning for the small evaluation set.

On the topic of evalautuon, it is limited to a narrow task suite. Specifically, the paper tests only three manipulation tasks (Fold Cloth, Clean Desk, Throw Bottle) on a single hardware platform in the real world, making the reported results both not reliable and not reproducible. In addition, the authors introduce a “behavior score” metric in addition to “success rate” but never define how it’s computed or normalized.

In the experimental evaluation of video generation quality the proposed approach outperforms CosmosTransfer in terms of multi-view consistency, which is not surprising given CosmosTransfer is a single view model.

Experiments demonstrate advantages of the proposed data augmentation approach during policy learning  compared to CosmosTransfer and not using data augmentation at all, but again very few details are provided. We don’t even know how many episodes were used for training. Instead of starting from a fixed set of episodes and seeing how much benefits can augmentation bring, the authors only evaluate how replacing some of the real samples with augmented ones can help, which is not the most realistic scenario.

Moreover, using CosmosTransfer as the only baseline is a bit misleading as it does not support multi-view inference, making it a weak baseline for evaluation in a multi-view setting. The paper’s framing (“DreamTransfer achieves consistent and substantial improvements”) thus overstates novelty, since the baseline was not capable of leveraging multiple camera inputs at all.

The reported 13% gain when integrating AdaMix with DreamTransfer is meaningful but modest.  Without a direct comparison to simpler baselines (e.g., reweighting by loss magnitude), and clear ablations on the metrics used for scoring, it’s hard to claim that AdaMix’s improvement is due to its design rather than generic hard-sample mining.

**Questions:**

Please address the issues with the missing details and inadequate evaluation described above. Specifically, please provide an evaluation on an established sim benchmark (e.g. RoboCasa) that shows that adding augmented samples to the existing set of demonstrations can improve the existing methods' performance of the benchmark.

---

### Official Review · Reviewer_DjYN · 2025-10-30

**Soundness:** 3
**Presentation:** 2
**Contribution:** 1
**Rating:** 2
**Confidence:** 3

**Summary:**

The paper proposes the EMMA framework, whose main emphasis is on generating geometrically/semantically consistent world-to-world transfer video. EMMA consists of two components: 1) DreamTransfer and 2) AdaMix.

DreamTransfer is the video generative backbone EMMA. DreamTransfer is an architectural improvement of existing method, RoboTransfer. Besides architectural improvement, what sets EMMA appart from existing method approaches is the addition of AdaMix. AdaMix is a kind of automated data curation/curriculum pipeline that adaptively filters out bad (=3d inconsistent) samples for better training. The authors propose several consistency metrics such as depth/multiview/prompt consistency to assess/filter/reweight the samples.

Experiments are designed to validate the core claims: 1) DreamTransfer excels in terms of geometric/semantic consistency, 2) Real world VLA adaptation, and 3) assessment on the impact of adaptive data reweighting/mixing ratio using AdaMix.

**Strengths:**

Benchmark results and confirm that DreamTransfer significantly improves geometric/semantic consistency over RoboTransfer and Cosmos-Transfer1. Combined with AdaMix, the EMMA achieves roughly 10-20% improvement in success rates accross three real-world tasks.

**Weaknesses:**

1. It is unclear from the paper why DreamTransfer achieves more consistent transfer. Specifically, it is hard to understand the main difference between DreamTransfer and RoboTransfer. If I understand correctly, the usage of DiT is the main difference. Although the advantage is validated by the experimental results, the contribution itself is rather incremental, given that a body of literature on 3D consistent generation already exists.

2. The usage of AdaMix is the core methodological novelty of this paper. While methodologically sound, the experimental results are statistically insignificant to support its claimed benefit. 20 trials on 3 task and an average of 15% improvement isn't significant enough to support the claim that AdaMix can provide substantial benefit on various real-world tasks that outweighs the added complexity of the pipeline.

**Questions:**

Why DreamTransfer is more geometrically consistent than RoboTransfer? What is its main advantage over numerous existing 3D/4D consistent video generative models? Why can't they be applied to EMMA framework?

---

### Official Review · Reviewer_ThHt · 2025-10-31

**Soundness:** 3
**Presentation:** 3
**Contribution:** 3
**Rating:** 4
**Confidence:** 3

**Summary:**

This work presents EMMA, a system designed to improve VLA policy generalization by pairing a generative multi-view video engine (DreamTransfer) with an adaptive sampling strategy (AdaMix). DreamTransfer generates geometry-consistent robot manipulation videos with text-controlled appearance modifications, supporting both real-world and simulation inputs. AdaMix further refines training by identifying difficult trajectories and up-weighting them during learning. Overall, the method aims to increase visual diversity and emphasize challenging samples, leading to improved robustness and zero-shot generalization in real-world manipulation tasks.

**Strengths:**

1. The idea of generating realistic videos from depth maps combined with textual descriptions, while achieving generalization at the visual level, is very interesting. This approach has strong potential, as it leverages both geometric cues and semantic information, which could lead to more robust world-model learning and scene understanding.

2. The baseline settings in this paper appear reasonable, and they effectively highlight the advantages of the proposed method.

**Weaknesses:**

1. The experimental section of the paper is somewhat weak. The real-world experiments involve limited task diversity, and the number of test trials is relatively small. This lack of breadth and repetition undermines the credibility and robustness of the experimental results. A richer evaluation protocol would help better demonstrate the method’s practical effectiveness.

2. The paper does not provide information on the computational cost associated with the different baselines. For world-model training, computational efficiency is extremely important—two models with similar architectures but trained for different durations can lead to significantly different conclusions. Reporting training time, GPU hours, and computational resources would strengthen the fairness and rigor of the comparison.

**Questions:**

It remains unclear whether the model is truly capable of generalizing to unseen environments. The paper would benefit from including both qualitative and quantitative results that showcase the model’s performance in novel scenarios. Demonstrating zero-shot or few-shot generalization across diverse settings would greatly enhance the claims regarding generalizability.

---

### Note · Authors · 2025-11-12

I have read and agree with the venue's withdrawal policy on behalf of myself and my co-authors.